# Complications of biliary stenting versus T-tube insertion after common bile duct exploration: A systematic review and meta-analysis

**Reno Rudiman**[1]*, **Ricarhdo Valentino Hanafi**[2], **Almawijaya**[1], **Freda Halim**[3]

**1** Division of Digestive Surgery, Department of General Surgery, School of Medicine, Universitas Padjadjaran, Hasan Sadikin General Hospital, Bandung, Indonesia, **2** Faculty of Medicine, Universitas Pelita Harapan, Tangerang, Indonesia, **3** Department of General Surgery, Faculty of Medicine, Universitas Pelita Harapan, Tangerang, Indonesia

* rudiman@unpad.ac.id

**Data Availability Statement:** Data are available from the following repository: https://doi.org/10. 5281/zenodo.7263085.

## Abstract

### Background

Complications following the insertion T-tube or stent after common bile duct exploration (CBDE) remain problematic in nowadays surgical era. Based on our knowledge, we did not find any meta-analysis intentionally evaluating the complications between both groups. At this moment, we aimed to analyze and compare both procedures' complications, efficacy, efficiency, and feasibility.

### Methods

We searched literature from four databases (EuroPMC, PubMed, Scopus, and Clinical-Trials.gov) up to June 2022 to compile the randomized controlled trials and pro-/retrospective cohort studies. Review Manager 5.4 was used to statistically analyze each outcome measured between biliary stenting and T-tube insertion.

### Results

Sixteen studies with 1,080 patients (534 biliary stents and 546 T-tube) were included for qualitative and quantitative analysis. The pooled risk ratio (RR) of the overall postoperative complications rate was significantly lower in the biliary stent group compared to the T-tube group 0.43 [95% confidence interval (CI) 0.23–0.80, $p = 0.007$]. In terms of the operation time, length of hospital stay, and readmission rate was also decreased in stenting as biliary drainage over T-tube placement 1.02 minutes [95% CI -1.53, -0.52, $p < 0.0001$], 1.96 days [95% CI -2.63, -1.29, $p < 0.00001$], and RR 0.39 [95% CI 0.15–0.97, $p = 0.04$], respectively.

### Conclusions

Stenting as biliary drainage after CBDE was superior to T-tube insertion. A shorter operation time and hospital stay in biliary drainage resulted in a lower overall postoperative complication rate. Other influences, including the complexity and shorter learning curve, might also affect the superiority of biliary stenting.

**Funding:** The author(s) received no specific funding for this work.

**Competing interests:** The authors have declared that no competing interests exist.

## Introduction

Common bile duct exploration (CBDE), either open or laparoscopically, is a procedure to treat and prevent further complications such as cholangitis, pancreatitis, and obstructive jaundice due to common bile duct stones. Choledocholithiasis may be a silent disease that, over time, generates further troubles [1]. Up to 20% of patients with cholelithiasis were classified as choledocholithiasis [2].

Following the CBDE, biliary drainage with T-tube or biliary stent is inserted within the common bile duct to prevent biliary leakage, stenosis, and other choledocholithiasis [3, 4]. This T-tube insertion was commonly performed before the endoscopic retrograde cholangiopancreatography (ERCP) era, the less invasive and renowned procedure in which safer internal biliary drainage could be achieved [5, 6]. Conversely, ERCP was not frequently utilized in developing countries due to the inequality of healthcare services.

The pioneer of T-tube drainage following choledochotomy was Deaver in 1904 [7]. Up to this day, T-tube has four types of material: polyvinyl chloride (PVC), red rubber, silicone, and latex. The latex T-tube is the most commonly used due to its superiority in complications, inflammatory reaction, and low bile sedimentation within the lumen. Generally, T-tube may be removed within 7 to 10 days if all clinical assessments improve [8]. It was a standard technique up to two decades ago due to its high association with numerous complications such as leakage, biliary peritonitis, T-tube dislodgment, vascular injury, and fistula. This procedure is a complex technique and requires an experienced operator to minimize the complications. Further damages, such as biliary leakage and biliary peritonitis, might occur when the T-tube is taken [9–11].

A biliary stent is a cylindrical device (plastic or metal) frequently used to establish the patency of the bile duct. Plastic and self-expanding metal stents (SEMs) showed a similar success rate in relieving bile duct stones and bile leakage [12]. Additionally, biliary stent removal can be removed safely endoscopically, which minimizes additional complications [13]. The previous report demonstrated the biliary complications from T-tube do not differ in open and laparoscopic procedures (15.5% vs 13.8%, respectively), and those complications were more related to the placement and removal of the T-tube drainage itself [14, 15]. Due to these controversies among surgeons, a shift towards biliary stenting occurred, demonstrating better outcomes and lower complication rates [16–18]. Previous meta-analysis conducted by Yin *et al.* 2013 showed primary duct closure (PDC) alone was superior to combination with T-tube insertion [19]. Moreover, Jiang *et al.* 2019 also stated that PDC alone or PDC combined with internal drainage (stenting) was superior to PDC with T-tube insertion [20].

Based on our awareness, we did not find any meta-analysis that purposefully compared T-tube complications and biliary stent drainage after CBDE (open or laparoscopically). In developed countries, T-tube insertion and open CBDE were left behind a decade ago. However, many developing countries still undergo those procedures due to the limitations of healthcare facilities. Consequently, we aimed to analyze and compare both procedures' complications, efficacy, efficiency, and feasibility.

## Materials & methods

### Eligibility criteria

The inclusion criteria are framed with the PICOS formula. (1) Population: patients with common bile duct (CBD) exploration (either open or laparoscopic), (2) Intervention: patients who were receiving biliary stenting, (3) Comparison: patients who were receiving T-tube drainage, (4) Primary outcomes: biliary leakage, wound infection, pancreatitis, overall postoperative

complications; secondary outcomes: operation time, blood loss volume, length of hospital stay, readmission rate, and reoperation rate, and lastly, (5) Study: randomized or non-randomized controlled trials, cohort, case-control, cross-sectional, and case-series.

We excluded any study with a population under 18 years old, pregnant women, and no control group. Non-English and Non-original studies, including review articles, editorials, letters to the editor, correspondences, and conference abstracts were also exclusions. This study concurs with the preferred reporting items for systematic reviews and meta-analyses (PRISMA) guidelines and is registered in the international prospective register of systematic reviews (PROSPERO) CRD42022360170.

### Information sources

We conducted literature searches in four databases (EuroPMC, PubMed, Scopus, and Clinical-Trials.gov) until June 2022 with English-language restrictions.

### Search strategy

The following keywords of this study are "(biliary OR bile duct OR common bile duct OR CBD) AND (exploration OR surgery OR open-choledochotomy OR laparoscopic OR decompression) AND (stent OR stenting) AND (T-tube)".

### Selection process

Initial screening of titles and abstracts with Covidence software (https://covidence.org) by two researchers (RR, RVH) and any discrepancies were solved with the third researcher (FH) discussion and consultation.

### Data collection process

Full-text screening and data extractions (author details, patient's characteristics, study design, outcomes measured) were performed by RR and RVH independently. Furthermore, FH merged and validated the two extracted data. Any incongruity was resolved through discussion with the third researcher.

### Data items

The outcomes were operation time (from the first incision until wound closure), blood loss volume (total blood loss during operation), length of hospital stay (as defined by each study), biliary leakage (definition of leakage by each study), wound infection (definition of infection by each study), pancreatitis (as defined by each study), postoperative complications (determined by each study), readmission rate (hospitalized due to specific reason after operation), and reoperation rate (second operation due to particular reason after the procedure).

### Study risk of bias assessment

We used the Newcastle-Ottawa scale (NOS) to evaluate the quality of case-control and cohort studies. The assessment reviews each study's comparability, selection, and outcome [21]. Meanwhile, a modified Jadad scale is a procedure to assess the quality of included clinical trial studies. It evaluates the four domains: the randomization process, allocation concealment, blindness, and withdrawals and drop-outs of the reported results [22]. Two researchers (RR, RVH) evaluated those risk of bias assessments, and a third researcher (FH) interceded with any divergence.

### Effect measures

The effect measures used in synthesizing results depend on the variable outcomes. We used a risk ratio (RR) with a 95% confidence interval (CI) for the dichotomous variable and standardized mean difference (SMD) and its standard deviations (SD) for the continuous variable. The summarized synthesis is presented in the forest plot.

### Synthesis methods

We utilized Review Manager 5.4 software for conducting meta-analysis. A Mantel-Haenszel formula with random-effect models was applied to calculate the RR and 95% CI. This formula allows to calculate an unconfounded, overall effect estimate of a given exposure for a specific outcome by combining the RR [23]. Meanwhile, an Inverse-Variance formula with random-effect models was used to assess the SMD and SD. This method involves a weighted average of the effect estimates from the separate studies, and the weight in each study is the inverse of the variance of the effect estimate [24]. We synthesized the heterogeneity using $I^2$ statistics. A 50% heterogeneity is considered substantial heterogeneity, which indicates half of the variability among effect sizes is due to significant heterogeneity among studies. Moreover, we applied Wan X *et al.* [25] formula to convert means and standard deviations (SD) for meta-analysis data synthesis.

### Publication bias

We employed funnel plots analysis to visualize the qualitative risk of publication bias.

## Results

### Study selection

Our initial databases search revealed 838 titles and abstracts. After removing the duplicates, screening titles, and abstracts, 63 publications were eligible for further full-text assessment. The final 16 full-text publications were qualified for qualitative and quantitative synthesis [26–41]. However, we excluded 47 publications for several reasons, including 27 articles that did not include biliary stenting, 12 articles that did not have any comparison or control group, five articles that were not primary research, and three articles that were not English papers. The summarized PRISMA flow chart of study selection is presented in Fig 1.

### Study characteristics

The final 16 full-text publications included 1,080 patients (534 biliary stents and 546 T-tube), published between 2004–2022. The study design were five prospective RCT studies, eight retrospective cohort studies, and three prospective cohort studies. Most of the included study participants received 7 or 10-Fr biliary stents. The detailed study characteristics are summarized in Table 1.

### Risk of bias in studies

The results of NOS indicated all the included studies had good quality (Table 2). The modified Jadad scale revealed one RCT was graded as a high-quality study; on the other hand, four RCTs were graded as moderate-quality studies (Table 3).

### Biliary stent vs. T-tube

**Bile leakage.** The pooled analysis from 12 studies (n = 811) demonstrated no differences between biliary stenting and T-tube drainage in bile leakage incidence [RR 0.67 (95% CI 0.36–1.27), $p$ = 0.22, $I^2$ = 0%, random-effect modeling] (Fig 2A).

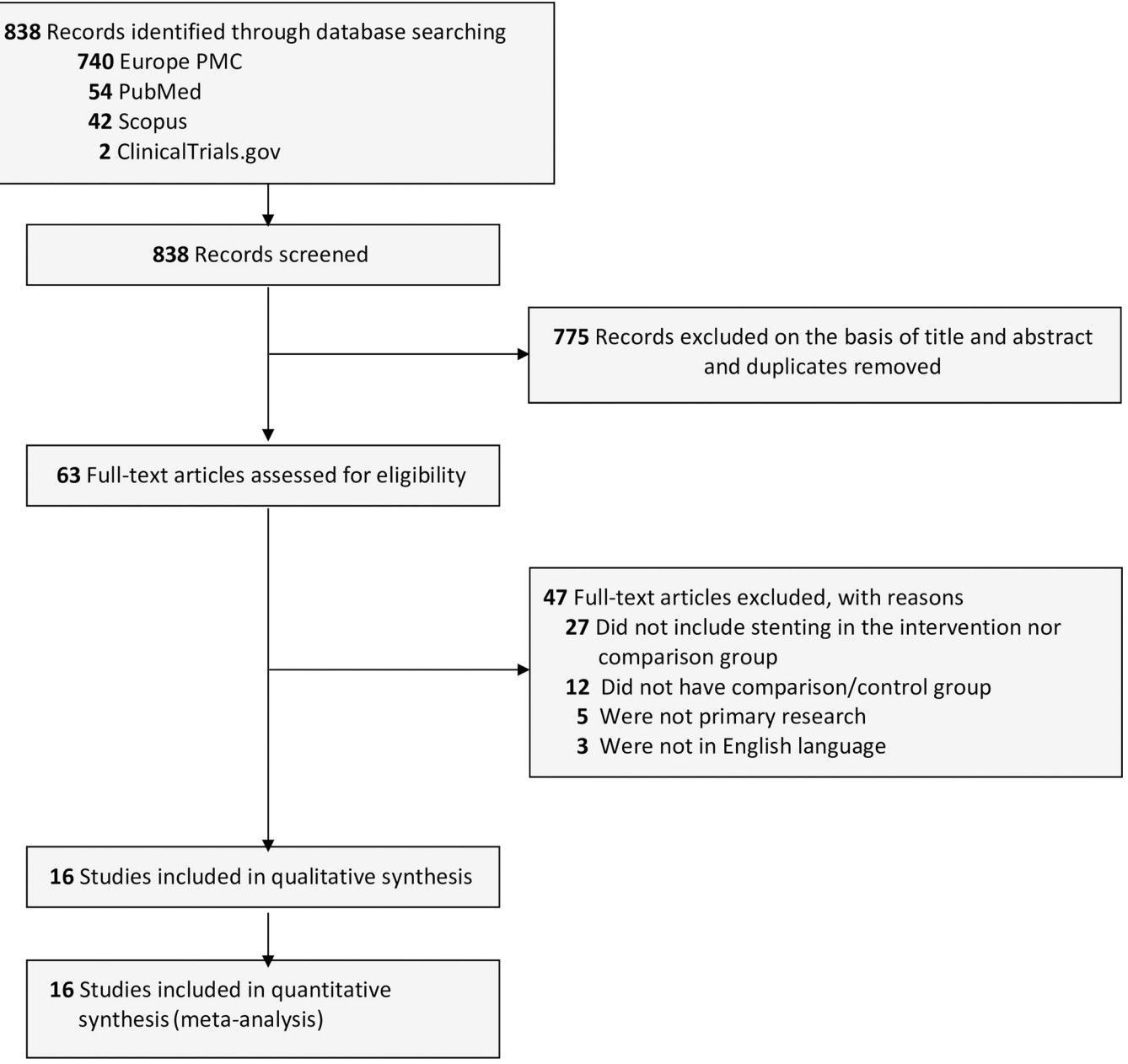

**Fig 1. The PRISMA flowchart of the studies selection process.**

**Wound infection.** In the pooled analysis of six studies (n = 269), biliary stenting and T-tube drainage after CBD exploration did not demonstrate any differences in wound infection outcome [RR 0.37 (95% CI 0.12–1.13), $p = 0.08$, $I^2 = 12\%$, random-effect modeling] (Fig 2B).

**Pancreatitis.** In the pooled analysis of eight studies (n = 363), the pancreatitis incidence after CBD exploration did not differ between biliary stenting and T-tube drainage [RR 0.55 (95% CI 0.13–2.28), $p = 0.41$, $I^2 = 41\%$, random-effect modeling] (Fig 2C).

**Overall postoperative complications.** In the pooled analysis of 16 studies (n = 1,080), the overall postoperative complications were lowered in the biliary stenting group than in the T-tube group after CBD exploration [RR 0.43 (95% CI 0.23–0.80), $p = 0.007$, $I^2 = 63\%$, random-effect modeling] (Fig 2D).

Table 1. Data characteristics of the included studies.

| First author, year | Sample size | | Design | Age (years) | | Gender (male/female) | | Stent used | Outcomes* |
|---|---|---|---|---|---|---|---|---|---|
| | Stent | T-tube | | Stent | T-tube | Stent | T-tube | | |
| Abdelkader AM et al. [26] 2018 | 25 | 25 | Prospective cohort | 41.2 ± 8.1 | 40.6 ± 7.5 | 11/14 | 10/15 | 8 or 10 Fr Nelaton tube | 1,2,3,4,7 |
| Dietrich A et al. [27] 2014 | 13 | 35 | Retrospective cohort | 60.2 ± 12.5 | 64.1 ± 15 | 9/5 | 18/17 | 7-Fr plastic biliary stent (Cotton-Leung) | 1,3,6,7,9 |
| Gyalpo T et al. [28] 2018 | 20 | 20 | Prospective RCT | 51.1 ± 11.5 | 50.6 ± 12 | 4/16 | 3/17 | 7-Fr 10cm straight flap biliary stent (Indovasive, India) | 1,3,4,5,7,8,9 |
| Hassan AM et al. [29] 2017 | 15 | 9 | Retrospective cohort | 49.5 ± 6.6 | 49.7 ± 6 | 15/6 | 5/4 | 7-Fr plastic stent | 1,3,6,7,9 |
| Isla AM et al. [30] 2004 | 21 | 32 | Retrospective cohort | 54 ± 14 | 65.1 ± 17.7 | 5/16 | 9/23 | 10-Fr biliary stent (Olympus) | 1,3,4,7 |
| Kwon SU et al. [31] 2011 | 23 | 29 | Retrospective cohort | 73 ± 9.1 | 69.7 ± 10.5 | 5/18 | 12/17 | Not mentioned in details | 1,3,4,7 |
| Xiao L et al. [32] 2018 | 120 | 116 | Retrospective cohort | 53.8 ± 12.8 | 51 ± 12.5 | 52/68 | 54/62 | 7 to 10-Fr biliary stent (ENDO-FLEX GmbH Company) | 1,3,4,7,8 |
| Lyon M et al. [33] 2015 | 82 | 34 | Prospective cohort | 43.8 ± 10 | 45.9 ± 12.5 | 21/61 | 6/28 | 7- Fr straight or duodenal curve biliary Diagmed™ stent | 3,7,9 |
| Mangla V et al. [34] 2012 | 31 | 29 | Prospective RCT | 46.8 ± 14.8 | 47.1 ± 12.3 | 9/22 | 5/24 | 7 or 10-Fr, 10-cm straight flap biliary stent (Devon, Agra, India) | 1,2,3,4,5,7 |
| Abd El Wahab A et al. [35] 2022 | 10 | 10 | Prospective cohort | 48.7 ± 4.9 | 53.1 ± 6.9 | N/A | N/A | Not described in detail | 1,2,3,4,5,6,7 |
| Martinez-Baena D et al. [36] 2013 | 28 | 47 | Retrospective cohort | 61.9 ± 15.2 | 61.1 ± 16.9 | 9/19 | 24/23 | Biliary stenting (Flexima®, Boston Scientific Corporation) | 3,4,6,7,8,9 |
| Mir IS et al. [37] 2016 | 11 | 20 | Retrospective cohort | N/A | N/A | N/A | N/A | 7 or 10-Fr biliary stent | 1,3,4,5,6,7 |
| Parra-Membrives P et al. [38] 2017 | 58 | 52 | Retrospective cohort | 63.2 ± 15.7 | 60.7 ± 15.2 | 18/40 | 25/27 | 6 to 8.5-Fr plastic biliary stent | 3,4,7,8,9 |
| Perez G et al.[39] 2005 | 37 | 44 | Prospective RCT | 56.9 ± 18.7 | 55 ± 16.3 | 34/10 | 28/9 | 5 or 6-Fr biliary stent (Zimmon, Wilson-Cook Medical Winston-Salem, NC, USA) | 3,5,6,7,8,9 |
| Redwan AA et al. [40] 2006 | 18 | 19 | Prospective RCT | 46 ± 12 | 46 ± 12 | N/A | N/A | 8.5-Fr biliary stent | 1,3,4,5,6,7 |
| Xu Y et al. [41] 2013 | 22 | 25 | Prospective RCT | 61.3 ± 21.8 | 62.1 ± 15.5 | 9/13 | 14/11 | 8.5 to 10-Fr biliary stent (Leadgem Medical Co., serial number FY0825) | 1,2,3,4,6,7,9 |

*Outcomes: 1 = operation time; 2 = blood loss volume; 3 = length of hospital stay; 4 = bile leakage; 5 = wound infection; 6 = pancreatitis; 7 = post-operative complications; 8 = readmission rate; 9 = reoperations rate

**Operation time.** In the pooled analysis from 12 studies (n = 698), biliary stenting as the drainage method after CBD exploration demonstrated a shorter operative time than T-tube [SMD -1.02 (95% CI -1.53, -0.52), $p < 0.0001$, $I^2$ = 88%, random-effect modeling] (Fig 2E).

**Blood loss volume.** The pooled analysis from four studies (n = 177) demonstrated both biliary stenting and T-tube drainage after CBD exploration did not differ in terms of blood loss volume [SMD -0.45 (95% CI -1.11, 0.21), $p = 0.18$, $I^2$ = 77%, random-effect modeling] (Fig 3A).

**Length of hospital stay.** In the pooled analysis from 16 studies (n = 1,080), the length of hospital stay was shorter in the biliary stenting method than T-tube after CBD exploration [SMD -1.96 (95% CI -2.63, -1.29), $p < 0.00001$, $I^2$ = 95%, random-effect modeling] (Fig 3B).

**Readmission rate.** In the pooled analysis of four studies (n = 306), the biliary stenting group exhibited a lower hospital readmission rate compared with the T-tube group [RR 0.39 (95% CI 0.15–0.97), $p = 0.04$, $I^2$ = 0%, random-effect modeling] (Fig 3C).

**Table 2. Newcastle-Ottawa quality assessment of observational studies.**

| First author, year | Study design | Selection[a] | Comparability[b] | Outcome[c] | Total score | Result |
|---|---|---|---|---|---|---|
| Abdelkader AM et al. [26] 2018 | Cohort | *** | ** | ** | 7 | Good |
| Dietrich A et al. [27] 2014 | Cohort | *** | ** | *** | 8 | Good |
| Hassan AM et al. [29] 2017 | Cohort | *** | ** | ** | 7 | Good |
| Isla AM et al. [30] 2004 | Cohort | *** | ** | *** | 8 | Good |
| Kwon SU et al. [31] 2011 | Cohort | *** | ** | ** | 7 | Good |
| Xiao L et al. [32] 2018 | Cohort | *** | ** | *** | 8 | Good |
| Lyon M et al. [33] 2015 | Cohort | *** | ** | ** | 7 | Good |
| Abd El Wahab A et al. [35] 2022 | Cohort | *** | ** | ** | 7 | Good |
| Martinez-Baena D et al. [36] 2013 | Cohort | *** | ** | ** | 7 | Good |
| Mir IS et al. [37] 2016 | Cohort | *** | ** | ** | 7 | Good |
| Parra-Membrives P et al. [38] 2017 | Cohort | *** | ** | *** | 8 | Good |

[a](1) representativeness of the exposed cohort; (2) selection of the non-exposed cohort; (3) ascertainment of exposure; (4) demonstration that outcome of interest was not present at the start of the study

[b](1) comparability of cohorts based on design or analysis (maximum two stars)

[c](1) assessment of outcome; (2) was follow-up long enough for outcomes to occur; (3) adequacy of follow-up of cohorts

**Reoperation rate.** In the pooled analysis of nine studies (n = 677), the reoperation rate between the biliary stenting and T-tube groups revealed no differences significantly [RR 0.59 (95% CI 0.25–1.42), $p$ = 0.24, $I^2$ = 0%, random-effect modeling] (Fig 3D).

## Publication bias

We assessed the risk of bias using Funnel plot analysis for each synthesis. Our analysis illustrated a relatively symmetrical inverted plot for all syntheses. Thus, our Funnel plot analysis indicates no publication bias in this study (Fig 4A–4I).

## Discussion

Based on the included studies, T-tube insertion as biliary drainage after CBDE has been unpopular in the modern surgical era. Ten out of 16 studies were conducted before 2017. This trend is shifting towards the biliary stenting era in developed countries. A similar meta-analysis had been conducted by Yin *et al.* and Jiang *et al.*; nonetheless, our study did not discern the operative procedure (open or laparoscopic) because we aimed to solely analyze the safety and feasibility between T-tube and biliary stent insertion after CBDE [19, 20].

**Table 3. Quality appraisal of studies included in the meta-analysis using the Jadad scale assessment.**

| Study | Random allocation | Concealment schemes | Blinding | Withdrawals and Drop-out | Total score | Interpretation |
|---|---|---|---|---|---|---|
| Gyalpo T et al. [28] 2018 | 2 | 0 | 1 | 1 | 4 | Moderate quality |
| Mangla V et al. [34] 2012 | 2 | 2 | 2 | 1 | 7 | High quality |
| Perez G et al. [39] 2005 | 1 | 1 | 1 | 0 | 3 | Moderate quality |
| Redwan AA et al. [40] 2006 | 1 | 1 | 1 | 1 | 4 | Moderate quality |
| Xu Y et al. [41] 2013 | 1 | 1 | 1 | 0 | 3 | Moderate quality |

Interpretation:

More than 4 points : High quality

3–4 points : Moderate quality

Less than 3 points : Low quality

**A.**

| Study or Subgroup | Stent Events | Total | T-tube Events | Total | Weight | Risk Ratio M-H, Random, 95% CI |
|---|---|---|---|---|---|---|
| Abdelkader AM et al. 2018 | 0 | 25 | 1 | 25 | 4.0% | 0.33 [0.01, 7.81] |
| Gyalpo T et al. 2018 | 0 | 20 | 1 | 20 | 4.0% | 0.33 [0.01, 7.72] |
| Isla AM et al. 2004 | 0 | 21 | 3 | 32 | 4.7% | 0.21 [0.01, 3.95] |
| Kwon SU et al. 2011 | 1 | 23 | 1 | 29 | 5.4% | 1.26 [0.08, 19.09] |
| Lin-Kang X et al. 2018 | 0 | 120 | 2 | 116 | 4.4% | 0.19 [0.01, 3.99] |
| Mangla V et al. 2012 | 1 | 31 | 2 | 29 | 7.3% | 0.47 [0.04, 4.89] |
| Mansour AESAE et al. 2022 | 0 | 10 | 3 | 10 | 4.9% | 0.14 [0.01, 2.45] |
| Martinez-Baena D et al. 2013 | 4 | 28 | 2 | 47 | 15.0% | 3.36 [0.66, 17.16] |
| Mir IS et al. 2016 | 0 | 11 | 1 | 20 | 4.1% | 0.58 [0.03, 13.22] |
| Parra-Membrives P et al. 2017 | 6 | 58 | 7 | 52 | 38.1% | 0.77 [0.28, 2.14] |
| Redwan AA et al. 2006 | 0 | 18 | 1 | 19 | 4.1% | 0.35 [0.02, 8.09] |
| Xu Y et al. 2016 | 0 | 22 | 1 | 25 | 4.0% | 0.38 [0.02, 8.80] |
| **Total (95% CI)** | | **387** | | **424** | **100.0%** | **0.67 [0.36, 1.27]** |
| Total events | 12 | | 25 | | | |

Heterogeneity: Tau² = 0.00; Chi² = 7.24, df = 11 (P = 0.78); I² = 0%
Test for overall effect: Z = 1.22 (P = 0.22)

**B.**

| Study or Subgroup | Stent Events | Total | T-tube Events | Total | Weight | Risk Ratio M-H, Random, 95% CI |
|---|---|---|---|---|---|---|
| Gyalpo T et al. 2018 | 1 | 20 | 11 | 20 | 27.3% | 0.09 [0.01, 0.64] |
| Mangla V et al. 2012 | 1 | 31 | 0 | 29 | 11.6% | 2.81 [0.12, 66.40] |
| Mansour AESAE et al. 2022 | 1 | 10 | 4 | 10 | 26.0% | 0.25 [0.01, 1.86] |
| Mir IS et al. 2016 | 0 | 11 | 0 | 20 | | Not estimable |
| Perez G et al. 2005 | 1 | 37 | 2 | 44 | 19.7% | 0.59 [0.06, 6.30] |
| Redwan AA et al. 2006 | 1 | 18 | 1 | 19 | 15.5% | 1.06 [0.07, 15.64] |
| **Total (95% CI)** | | **127** | | **142** | **100.0%** | **0.37 [0.12, 1.13]** |
| Total events | 5 | | 18 | | | |

Heterogeneity: Tau² = 0.19; Chi² = 4.54, df = 4 (P = 0.34); I² = 12%
Test for overall effect: Z = 1.74 (P = 0.08)

**C.**

| Study or Subgroup | Stent Events | Total | T-tube Events | Total | Weight | Risk Ratio M-H, Random, 95% CI |
|---|---|---|---|---|---|---|
| Dietrich A et al. 2014 | 0 | 35 | 2 | 13 | 13.8% | 0.08 [0.00, 1.52] |
| Hassan AM et al. 2017 | 0 | 15 | 3 | 9 | 14.5% | 0.09 [0.01, 1.55] |
| Mansour AESAE et al. 2022 | 0 | 10 | 5 | 10 | 15.0% | 0.09 [0.01, 1.45] |
| Martinez-Baena D et al. 2013 | 0 | 28 | 1 | 47 | 12.7% | 0.55 [0.02, 13.10] |
| Mir IS et al. 2016 | 2 | 11 | 1 | 20 | 18.4% | 3.64 [0.37, 35.73] |
| Perez G et al. 2005 | 1 | 37 | 0 | 44 | 12.7% | 3.55 [0.15, 84.69] |
| Redwan AA et al. 2006 | 1 | 18 | 0 | 19 | 12.9% | 3.16 [0.14, 72.84] |
| Xu Y et al. 2016 | 0 | 22 | 0 | 25 | | Not estimable |
| **Total (95% CI)** | | **176** | | **187** | **100.0%** | **0.55 [0.13, 2.28]** |
| Total events | 4 | | 12 | | | |

Heterogeneity: Tau² = 1.48; Chi² = 10.14, df = 6 (P = 0.12); I² = 41%
Test for overall effect: Z = 0.82 (P = 0.41)

**D.**

| Study or Subgroup | Stent Events | Total | T-tube Events | Total | Weight | Risk Ratio M-H, Random, 95% CI |
|---|---|---|---|---|---|---|
| Abdelkader AM et al. 2018 | 0 | 25 | 1 | 25 | 2.9% | 0.33 [0.01, 7.81] |
| Dietrich A et al. 2014 | 1 | 35 | 5 | 13 | 5.1% | 0.07 [0.01, 0.58] |
| Gyalpo T et al. 2018 | 2 | 20 | 18 | 20 | 7.8% | 0.11 [0.03, 0.42] |
| Hassan AM et al. 2017 | 1 | 15 | 5 | 9 | 5.4% | 0.12 [0.02, 0.87] |
| Isla AM et al. 2004 | 0 | 21 | 6 | 32 | 3.4% | 0.12 [0.01, 1.95] |
| Kwon SU et al. 2011 | 5 | 23 | 2 | 29 | 6.9% | 3.15 [0.67, 14.79] |
| Lin-Kang X et al. 2018 | 0 | 120 | 3 | 116 | 3.2% | 0.14 [0.01, 2.65] |
| Lyon M et al. 2015 | 0 | 82 | 4 | 34 | 3.3% | 0.05 [0.00, 0.85] |
| Mangla V et al. 2012 | 2 | 31 | 5 | 29 | 6.8% | 0.37 [0.08, 1.78] |
| Mansour AESAE et al. 2022 | 1 | 10 | 10 | 10 | 7.1% | 0.14 [0.03, 0.64] |
| Martinez-Baena D et al. 2013 | 10 | 28 | 15 | 47 | 10.7% | 1.12 [0.58, 2.14] |
| Mir IS et al. 2016 | 4 | 11 | 4 | 20 | 8.4% | 1.82 [0.56, 5.88] |
| Parra-Membrives P et al. 2017 | 16 | 58 | 14 | 52 | 10.9% | 1.02 [0.56, 1.89] |
| Perez G et al. 2005 | 4 | 37 | 13 | 44 | 9.0% | 0.37 [0.13, 1.03] |
| Redwan AA et al. 2006 | 3 | 18 | 2 | 19 | 6.4% | 1.58 [0.30, 8.40] |
| Xu Y et al. 2016 | 0 | 22 | 1 | 25 | 2.9% | 0.38 [0.02, 8.80] |
| **Total (95% CI)** | | **556** | | **524** | **100.0%** | **0.43 [0.23, 0.80]** |
| Total events | 49 | | 108 | | | |

Heterogeneity: Tau² = 0.80; Chi² = 40.96, df = 15 (P = 0.0003); I² = 63%
Test for overall effect: Z = 2.68 (P = 0.007)

**E.**

| Study or Subgroup | Stent Mean | SD | Total | T-tube Mean | SD | Total | Weight | Std. Mean Difference IV, Random, 95% CI |
|---|---|---|---|---|---|---|---|---|
| Abdelkader AM et al. 2018 | 65 | 20.5 | 25 | 90 | 17.32 | 25 | 8.7% | -1.30 [-1.91, -0.68] |
| Dietrich A et al. 2014 | 150 | 30 | 35 | 186.5 | 30 | 13 | 8.5% | -1.20 [-1.88, -0.51] |
| Gyalpo T et al. 2018 | 121.6 | 17.86 | 20 | 136.2 | 25.28 | 20 | 8.6% | -0.65 [-1.29, -0.02] |
| Hassan AM et al. 2017 | 146.73 | 12.34 | 15 | 191.33 | 10.13 | 9 | 5.6% | -3.72 [-5.13, -2.30] |
| Isla AM et al. 2004 | 102.5 | 7.5 | 21 | 112.5 | 5 | 32 | 8.6% | -1.61 [-2.25, -0.98] |
| Kwon SU et al. 2011 | 73 | 9.1 | 23 | 69.7 | 10.5 | 29 | 8.9% | 0.33 [-0.22, 0.88] |
| Lin-Kang X et al. 2018 | 106.88 | 5.91 | 120 | 106.71 | 5.19 | 116 | 9.8% | 0.03 [-0.22, 0.29] |
| Mangla V et al. 2012 | 139.19 | 18.26 | 31 | 161.1 | 19.21 | 29 | 8.9% | -1.15 [-1.70, -0.61] |
| Mansour AESAE et al. 2022 | 160.5 | 35.62 | 10 | 168 | 40.01 | 10 | 7.7% | -0.19 [-1.07, 0.69] |
| Mir IS et al. 2016 | 80 | 14.8 | 11 | 100 | 5 | 20 | 7.5% | -2.03 [-2.95, -1.12] |
| Redwan AA et al. 2006 | 117 | 17.5 | 18 | 128 | 22.5 | 19 | 8.6% | -0.53 [-1.19, 0.13] |
| Xu Y et al. 2016 | 93.75 | 18.75 | 22 | 145 | 45 | 25 | 8.6% | -1.43 [-2.08, -0.78] |
| **Total (95% CI)** | | | **351** | | | **347** | **100.0%** | **-1.02 [-1.53, -0.52]** |

Heterogeneity: Tau² = 0.65; Chi² = 91.08, df = 11 (P < 0.00001); I² = 88%
Test for overall effect: Z = 4.00 (P < 0.0001)

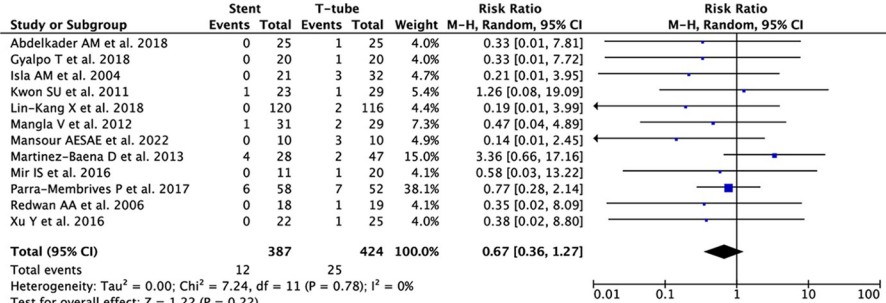

**Fig 2.** Forest plots of (A) bile leakage, (B) wound infection, (C) pancreatitis, (D) overall postoperative complications, and (E) operation time.

The previous meta-analysis stated that T-tube insertion yielded higher biliary complications [19, 20]. On the other hand, our analytical synthesis demonstrated a similar outcome in terms of biliary leakage (n = 881) and pancreatitis (n = 363) with [RR 0.67 (95% CI 0.36–1.27), $p$ = 0.22, $I^2$ = 0%, random-effect modeling] and [RR 0.55 (95% CI 0.13–2.28), $p$ = 0.41, $I^2$ = 41%, random-effect modeling], respectively. These contradictory findings could be the limitation of the previous meta-analysis, which underwent the procedure laparoscopically. Visual limitation during a laparoscopic procedure when the CBD is fragile, edematous, and prone to bleeding might escalate higher complication rate [42]. Therefore, biliary complications will increase when the T-tube is inserted with the laparoscopic technique in the previous meta-analysis.

Another concern from the patient's point of view is having a 'tube' on their tummy skin for weeks or months. Hygiene, patient activities, and external exposures might raise the possibility of wound infection [43]. Based on our latest analysis, T-tube drainage did not increase the wound infection incidence compared to biliary stenting [RR 0.37 (95% CI 0.12–1.13), $p$ = 0.08, $I^2$ = 12%, random-effect modeling]. Somehow, if we examine the overall postoperative complications (consisting of bile leakage, pancreatitis, wound infection, stricture, residual tone, bleeding, and bile duct injury), biliary stenting exhibited lower incidences [RR 0.43 (95% CI 0.23–0.80), $p$ = 0.007, $I^2$ = 63%, random-effect modeling] compared to T-tube group. This analysis ensured that the complexity of T-tube insertion affected postoperative complications.

Regarding efficiency, our pooled analysis from 12 studies (n = 698) showed operation time in biliary stenting as the drainage method after CBDE was shorter than the T-tube group. This finding is concurrent with the previous statement about the difficulty of T-tube insertion, and surgeons are prone to prefer biliary stenting as biliary drainage due to a shorter learning curve [44]. Shorter operation time and fewer complications mean a safer and preferable option for patients and operators.

Furthermore, we analyzed a blood loss volume that the previous meta-analysis had not conducted. This outcome is essential for patient safety and further complications. The pooled analysis from four studies (n = 177) exhibited no difference in blood loss volume between both groups [SMD -0.45 (95% CI -1.11, 0.21), $p$ = 0.18, $I^2$ = 77%, random-effect modeling].

Length of hospital stays and readmission rates are also crucial regarding potential nosocomial infections in the hospital [45, 46]. Our statistical analysis of hospital stay in 16 studies (n = 1,080) showed a shorter duration of hospital stay in the biliary stent group [SMD -1.96 (95% CI -2.63, -1.29), $p < 0.00001$, $I^2$ = 95%, random-effect modeling] and the readmission rate analysis from four studies (n = 306) was also significantly lower in the biliary stent group compared to the T-tube group [RR 0.39 (95% CI 0.15–0.97), $p$ = 0.04, $I^2$ = 0%, random-effect modeling]. Our findings were equivalent to prior meta-analysis [19, 20].

Lastly, the reoperation rate did not significantly differ between both groups [RR 0.59 (95% CI 0.25–1.42), $p$ = 0.24, $I^2$ = 0%, random-effect modeling]. Even though the overall postoperative complications were higher in the T-tube group, the reoperation rate did not differ statistically. This finding suggested that the postoperative complications could be managed with nonoperative approaches.

We realize in today's era, a less invasive approach has been a favorable technique, and T-tube should be dropped off. We would like to emphasize to the physicians that biliary stent should be a preferred method due to its superiority in several aspects (shorter operation time,

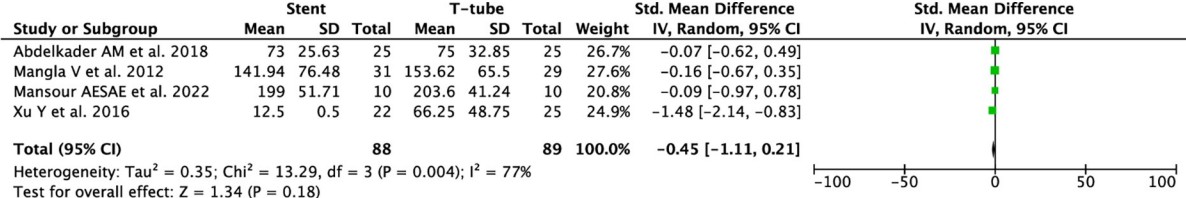

**Fig 3.** Forest plots of (A) blood loss volume, (B) length of hospital stay, (C) readmission rate, and (D) reoperation rate.

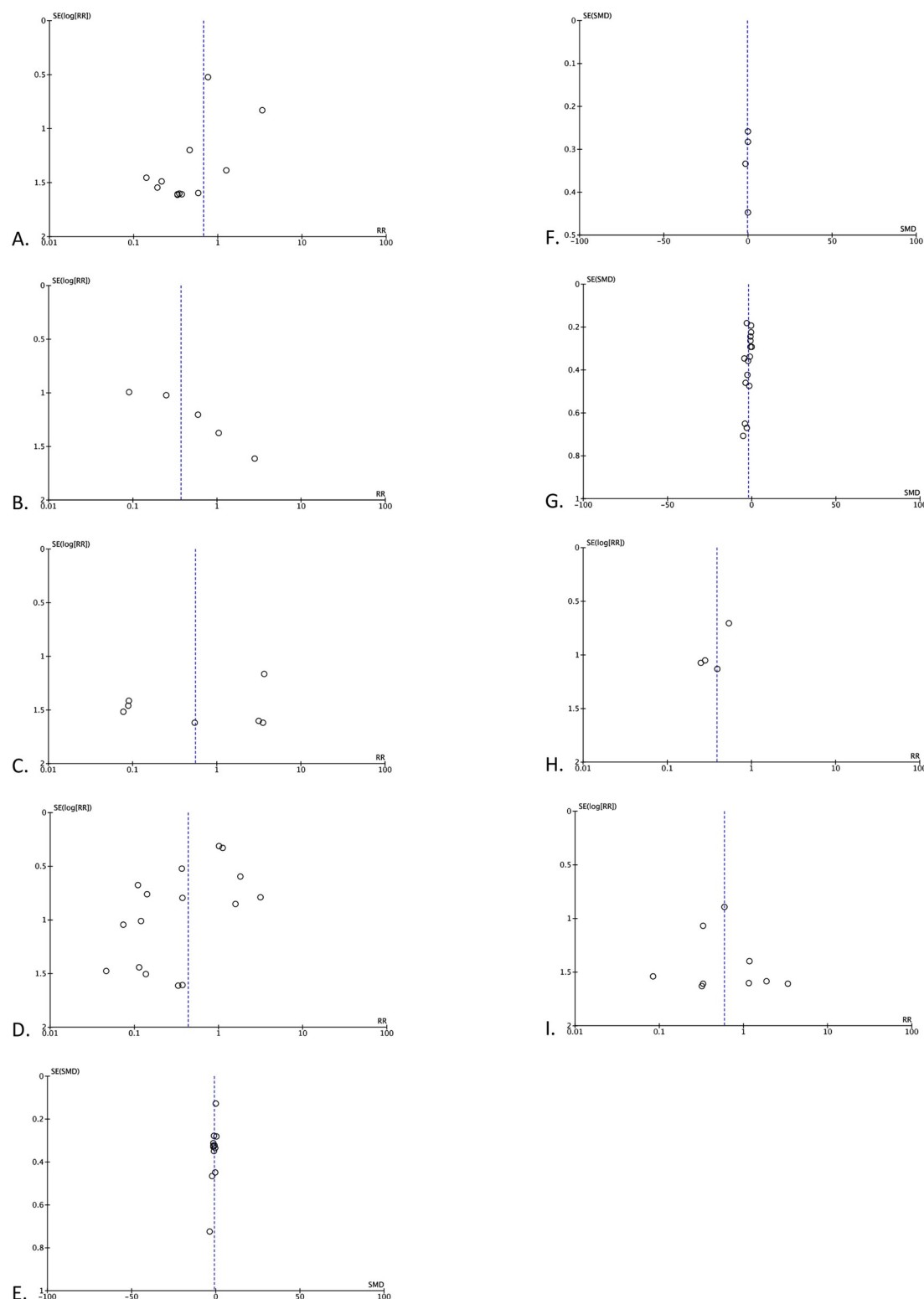

**Fig 4.** Funnel plots of (A) bile leakage, (B) wound infection, (C) pancreatitis, (D) overall postoperative complications, (E) operation time, (F) blood loss volume, (G) length of hospital stay, (H) readmission rate, and (I) reoperation rate.

shorter hospital stay, and lower overall complications rate). Despite the advancement of the biliary stent in the last decade, these two candidates still can be compared. Our included studies in the last five years assessed both procedures, and T-tube remains in the modern era that gradually fades over time.

There are some considerations regarding our meta-analysis limitations. First, only five out of 16 studies are categorized as RCTs. Second, ten studies were conducted before 2017. More updated and high-quality RCTs are needed to improve further meta-analysis. Third, high heterogeneity among the included studies might affect the overall outcomes. The contributions of high heterogeneity derive from data characteristics, the definition of each outcome, and outcome parameters. Lastly, English-only studies were also limiting our meta-analysis to extend further included studies.

## Conclusions

Stenting as biliary drainage after CBDE was superior to T-tube insertion. A shorter operation time and hospital stay in biliary drainage resulted in a lower overall postoperative complication rate. Other influences, including the complexity and shorter learning curve, might also affect the superiority of biliary stenting. We recommend stenting over T-tube insertion as biliary drainage; even though, T-tube cannot be left out entirely in developing countries with limited healthcare facilities.

## Supporting information

**S1 Checklist. PRISMA 2020 checklist.**
(PDF)

## Author Contributions

**Conceptualization:** Reno Rudiman, Ricarhdo Valentino Hanafi.

**Data curation:** Reno Rudiman, Ricarhdo Valentino Hanafi.

**Formal analysis:** Reno Rudiman, Ricarhdo Valentino Hanafi.

**Investigation:** Reno Rudiman, Ricarhdo Valentino Hanafi, Freda Halim.

**Methodology:** Reno Rudiman, Ricarhdo Valentino Hanafi, Freda Halim.

**Project administration:** Reno Rudiman.

**Resources:** Reno Rudiman.

**Software:** Reno Rudiman.

**Supervision:** Almawijaya.

**Validation:** Reno Rudiman, Ricarhdo Valentino Hanafi, Freda Halim.

**Visualization:** Reno Rudiman, Ricarhdo Valentino Hanafi, Freda Halim.

**Writing – original draft:** Reno Rudiman, Ricarhdo Valentino Hanafi, Almawijaya, Freda Halim.

**Writing – review & editing:** Reno Rudiman, Ricarhdo Valentino Hanafi, Almawijaya, Freda Halim.

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
