## [Decision Letter · Decision Letter 0]

28 Nov 2022

PONE-D-22-29837Complications of biliary stenting versus T-tube insertion after common bile duct exploration: A systematic review and meta-analysisPLOS ONE

Dear Dr. Reno Rudiman,

Thank you for submitting your manuscript to PLOS ONE. After careful consideration, we feel that it has merit but does not fully meet PLOS ONE’s publication criteria as it currently stands. Therefore, we invite you to submit a revised version of the manuscript that addresses the points raised during the review process.

We look forward to receiving your revised manuscript.

Kind regards,

Wenguo Cui, Ph.D

Academic Editor

PLOS ONE

2. Please include your tables as part of your main manuscript and remove the individual files. Please note that supplementary tables (should remain/ be uploaded) as separate "supporting information" files

Reviewers' comments:

Reviewer's Responses to Questions

**Comments to the Author**

1. Is the manuscript technically sound, and do the data support the conclusions?

Reviewer #1: Yes

Reviewer #2: Yes

2. Has the statistical analysis been performed appropriately and rigorously? 

Reviewer #1: Yes

Reviewer #2: Yes

3. Have the authors made all data underlying the findings in their manuscript fully available?

Reviewer #1: Yes

Reviewer #2: Yes

4. Is the manuscript presented in an intelligible fashion and written in standard English?

Reviewer #1: Yes

Reviewer #2: Yes

5. Review Comments to the Author

Reviewer #1: Author reviewed and did meta-analysis of 16 papers. These papers are published between 2004 – 2022 include 1,080 patients' (534 biliary stents and 546 T-tube) outcome of biliary stenting or T-tube insertion after common bile duct exploration operation. Author statistically analyzed the major complications rates, operation time, hospital stay length, etc. And author concluded stenting as biliary drainage after CBDE was superior to T-tube insertion.

Although the manuscript is drafted well and technically sounded good. The two candidates are just not comparable. Modern surgery are tend to be as less invasive as possible. While author also mentioned T-tube insertion already been dropped off by many places in the world and sooner or later it will not be used at all. On the other hand, the developing of stenting technology never stopped, stent itself even can be drug loaded or biodegradable.

Reviewer #2: This research article by Rudiman et al compared post T-tube or stent insertion with respect to procedure complications, efficacies and efficiencies. The authors analyzed 534 biliary stents and 546 T-tube patients and results showed that stenting after common bile duct exploration (CBDE) is superior to T-tube insertion with shorter operation time and hospital state as well as lower overall postoperative complication rate.

I believe that this manuscript is suitable for publication in Plos One. However, there are still some minor points that the authors need to address

1. all figures appear to be very blurry. Please upload high res figures.

2. It would be very informative to include more background information of stent and T-tube insertions in general.

2. Some sentences need to be described more clearly, for example, Page 16, “Thus, indicating no publication bias in this study…”, there was no subject in this sentence.

6. PLOS authors have the option to publish the peer review history of their article (what does this mean?). If published, this will include your full peer review and any attached files.

Reviewer #1: No

Reviewer #2: No

---

## [Author Response · Author response to Decision Letter 0]

7 Dec 2022

Dear Editor-in-Chief of PLoS ONE,

Thank you for allowing us to submit a revised draft of our manuscript. We appreciate the time and effort you and the reviewers have dedicated to providing your valuable feedback on our manuscript. We are grateful to the reviewers for their insightful comments on our paper. We have been able to incorporate changes to reflect most of the suggestions provided by the reviewers.

Here is a point-by-point response to the reviewers’ comments and concerns.

Reviewer 1

Author reviewed and did meta-analysis of 16 papers. These papers are published between 2004 – 2022 include 1,080 patients' (534 biliary stents and 546 T-tube) outcome of biliary stenting or T-tube insertion after common bile duct exploration operation. Author statistically analyzed the major complications rates, operation time, hospital stay length, etc. And author concluded stenting as biliary drainage after CBDE was superior to T-tube insertion.

Although the manuscript is drafted well and technically sounded good. The two candidates are just not comparable. Modern surgery are tend to be as less invasive as possible. While author also mentioned T-tube insertion already been dropped off by many places in the world and sooner or later it will not be used at all. On the other hand, the developing of stenting technology never stopped, stent itself even can be drug loaded or biodegradable.

Author’s comment: Thank you for your insightful comment. We agree that a less invasive approach has been a favorable technique in modern surgery. T-tube should be dropped off and shifted to minimally invasive method. Through this study, we would like to emphasize to the readers that biliary stent should be the preferred method due to its superiority in several aspects (shorter operation time, shorter hospital stay, and lower overall complications rate). The development of biliary stents has been quite remarkable in the last decade; however, these two candidates are feasible to be compared because in the last five years, our included studies still compared both methods and the limitation of healthcare services in developing countries force surgeons to choose T-tube as the initial option. With all deliberation, T-tube remains a realistic possibility without increasing patients’ mortality and morbidity in developing countries.

We also incorporated further reasons on writing this manuscript in the Discussion section.

Reviewer 2

1. All figures appear to be very blurry. Please upload high res figures.

Author’s comment: We have updated all figures using Prelight Analysis and Conversion Engine (PACE) digital diagnostic tool to ensure all figures meet PLOS requirements.

2. It would be very informative to include more background information of stent and T-tube insertions in general.

Author’s comment: Thank you for your consideration. In the Introduction section, we have incorporated additional background information on biliary stent and T-tube.

3. Some sentences need to be described more clearly, for example, Page 16, “Thus, indicating no publication bias in this study…”, there was no subject in this sentence.

Author’s comment: Thank you for your valuable feedback. We have completed that sentence. 

We look forward to hearing from you in due time regarding the submission, responses, further questions, and comments you may have.

Sincerely,

Reno Rudiman

---

## [Decision Letter · Decision Letter 1]

5 Jan 2023

PONE-D-22-29837R1Complications of biliary stenting versus T-tube insertion after common bile duct exploration: A systematic review and meta-analysisPLOS ONE

Dear Dr. Reno Rudiman,

Thank you for submitting your manuscript to PLOS ONE. After careful consideration, we feel that it has merit but does not fully meet PLOS ONE’s publication criteria as it currently stands. Therefore, we invite you to submit a revised version of the manuscript that addresses the points raised during the review process.

Please submit your revised manuscript by January 15, 2023.  If you will need more time than this to complete your revisions, please reply to this message or contact the journal office at plosone@plos.org. Please include the following items when submitting your revised manuscript:A rebuttal letter that responds to each point raised by the academic editor and reviewer(s). You should upload this letter as a separate file labeled 'Response to Reviewers'.A marked-up copy of your manuscript that highlights changes made to the original version. You should upload this as a separate file labeled 'Revised Manuscript with Track Changes'.An unmarked version of your revised paper without tracked changes. You should upload this as a separate file labeled 'Manuscript'.If applicable, we recommend that you deposit your laboratory protocols in protocols.io to enhance the reproducibility of your results. Protocols.io assigns your protocol its own identifier (DOI) so that it can be cited independently in the future. For instructions see: https://journals.plos.org/plosone/s/submission-guidelines#loc-laboratory-protocols. Additionally, PLOS ONE offers an option for publishing peer-reviewed Lab Protocol articles, which describe protocols hosted on protocols.io. Read more information on sharing protocols at https://plos.org/protocols?utm_medium=editorial-email&utm_source=authorletters&utm_campaign=protocols.

We look forward to receiving your revised manuscript.

Kind regards,

Wenguo Cui, Ph.D

Academic Editor

PLOS ONE

Journal Requirements:

Reviewers' comments:

Reviewer's Responses to Questions

**Comments to the Author**

1. If the authors have adequately addressed your comments raised in a previous round of review and you feel that this manuscript is now acceptable for publication, you may indicate that here to bypass the “Comments to the Author” section, enter your conflict of interest statement in the “Confidential to Editor” section, and submit your "Accept" recommendation.

Reviewer #1: (No Response)

Reviewer #2: All comments have been addressed

2. Is the manuscript technically sound, and do the data support the conclusions?

Reviewer #1: Yes

Reviewer #2: Yes

3. Has the statistical analysis been performed appropriately and rigorously? 

Reviewer #1: I Don't Know

Reviewer #2: Yes

4. Have the authors made all data underlying the findings in their manuscript fully available?

Reviewer #1: Yes

Reviewer #2: Yes

5. Is the manuscript presented in an intelligible fashion and written in standard English?

Reviewer #1: Yes

Reviewer #2: Yes

6. Review Comments to the Author

Reviewer #1: In this revision, author added more back ground information about T-tube insertion procedure and author also added explanation about how T- tube insertion compare to biliary stenting in discussion section. Manuscript is getting better but there are still have few more things I think author should make it more clear:

Author referred PRISMA guideline but Review Manager software has been used for conducting meta-analysis. Therefore, beside "The PRISMA flowchart of the studies selection process", how the Review Manager been configured and handled should be described in details since most of the data are generated by the Review Manager software in this paper.

Also, since this is a scientific paper doing meta-analysis. Statistical analysis in "Method" section should be explained with a lot more details. The manuscript mentioned 4 formulas in total. Please elaborate: Is there any result from PICOS formula? What is Mantel-Haenszel formula with random-effect models to calculate RR and CI? And what is Inverse-Variance formula with random-effect models that was used to assess the SMD and SD? Although formula to convert means

and standard deviations (SD) for meta-analysis data synthesis has a reference. How this formular been used in this manuscript needs more explanation.

Reviewer #2: (No Response)

7. PLOS authors have the option to publish the peer review history of their article (what does this mean?). If published, this will include your full peer review and any attached files.

Reviewer #1: No

Reviewer #2: No

---

## [Author Response · Author response to Decision Letter 1]

6 Jan 2023

Dear Editor-in-Chief of PLoS ONE,

Thank you for allowing us to submit a revised draft of our manuscript. We appreciate the time and effort you and the reviewers have dedicated to providing your valuable feedback on our manuscript. We are grateful to the reviewers for their insightful comments on our paper. We have been able to incorporate changes to reflect most of the suggestions provided by the reviewers.

Here is a point-by-point response to the reviewer’ comments and concerns.

Reviewer 1

Reviewer #1: In this revision, author added more back ground information about T-tube insertion procedure and author also added explanation about how T- tube insertion compare to biliary stenting in discussion section. Manuscript is getting better but there are still have few more things I think author should make it more clear: Author referred PRISMA guideline but Review Manager software has been used for conducting meta-analysis. Therefore, beside "The PRISMA flowchart of the studies selection process", how the Review Manager been configured and handled should be described in details since most of the data are generated by the Review Manager software in this paper. Also, since this is a scientific paper doing meta-analysis. Statistical analysis in "Method" section should be explained with a lot more details. The manuscript mentioned 4 formulas in total. Please elaborate: Is there any result from PICOS formula? What is Mantel-Haenszel formula with random-effect models to calculate RR and CI? And what is Inverse-Variance formula with random-effect models that was used to assess the SMD and SD? Although formula to convert means and standard deviations (SD) for meta-analysis data synthesis has a reference. How this formular been used in this manuscript needs more explanation.

Author’s comment: Thank you for your insightful suggestion.

1. Yes, there is a result from the PICOS formula, which we have elaborated in the Result Section and all the outcomes were statistically analysed

2. We have described further the Mantel-Haenszel and Inverse-Variance formula in the Synthesis Methods Section. 

We look forward to hearing from you regarding the submission, responses, questions, and comments you may have.

Sincerely,

Reno Rudiman

---

## [Editor Report · Decision Letter 2]

8 Jan 2023

Complications of biliary stenting versus T-tube insertion after common bile duct exploration: A systematic review and meta-analysis

PONE-D-22-29837R2

Dear Dr. Reno Rudiman,

We’re pleased to inform you that your manuscript has been judged scientifically suitable for publication and will be formally accepted for publication once it meets all outstanding technical requirements.

Kind regards,

Wenguo Cui, Ph.D

Academic Editor

PLOS ONE

---

## [Editor Report · Acceptance letter]

11 Jan 2023

PONE-D-22-29837R2 

Complications of biliary stenting versus T-tube insertion after common bile duct exploration: A systematic review and meta-analysis 

Dear Dr. Rudiman:

I'm pleased to inform you that your manuscript has been deemed suitable for publication in PLOS ONE. Congratulations! Your manuscript is now with our production department. 

Kind regards, 

on behalf of

Professor Wenguo Cui 

Academic Editor

PLOS ONE